# Development and Characterization of Novel Conductive Sensing Fibers for In Vivo Nerve Stimulation

**DOI:** 10.3390/s21227581

**Published:** 2021-11-15

**Authors:** Bertram Richter, Zachary Mace, Megan E. Hays, Santosh Adhikari, Huy Q. Pham, Robert J. Sclabassi, Benedict Kolber, Saigopalakrishna S. Yerneni, Phil Campbell, Boyle Cheng, Nestor Tomycz, Donald M. Whiting, Trung Q. Le, Toby L. Nelson, Saadyah Averick

**Affiliations:** 1System Department of Neurosurgery, Allegheny Health Network, Pittsburgh, PA 15212, USA; Bertramn.Richter@ahn.org (B.R.); zmace@cdi.com (Z.M.); bobs@cdi.com (R.J.S.); Boyle.CHENG@ahn.org (B.C.); Nestor.TOMYCZ@ahn.org (N.T.); Donald.Whiting@ahn.org (D.M.W.); 2Computational Diagnostics, Inc., Pittsburgh, PA 15213, USA; 3Department of Chemistry, Oklahoma State University, Stillwater, OK 74078, USA; mehays@ostatemail.okstate.edu (M.E.H.); santoshpeace50@gmail.com (S.A.); toby.nelson@okstate.edu (T.L.N.); 4Department of Biomedical Engineering, North Dakota State University, Fargo, ND 58102, USA; phamquochuy725@gmail.com; 5Department of Neuroscience, University of Texas at Dallas, Richardson, TX 75080, USA; Benedict.Kolber@UTDallas.edu; 6Department of Biomedical Engineering, Carnegie Mellon University, Pittsburgh, PA 15217, USA; gopalyerneni@gmail.com (S.S.Y.); pcampbel@cs.cmu.edu (P.C.); 7Department of Industrial and Manufacturing Engineering, North Dakota State University, Fargo, ND 58102, USA

**Keywords:** conductive sensing fiber, electrical probe, nerve stimulation

## Abstract

Advancements in electrode technologies to both stimulate and record the central nervous system’s electrical activities are enabling significant improvements in both the understanding and treatment of different neurological diseases. However, the current neural recording and stimulating electrodes are metallic, requiring invasive and damaging methods to interface with neural tissue. These electrodes may also degrade, resulting in additional invasive procedures. Furthermore, metal electrodes may cause nerve damage due to their inherent rigidity. This paper demonstrates that novel electrically conductive organic fibers (ECFs) can be used for direct nerve stimulation. The ECFs were prepared using a standard polyester material as the structural base, with a carbon nanotube ink applied to the surface as the electrical conductor. We report on three experiments: the first one to characterize the conductive properties of the ECFs; the second one to investigate the fiber cytotoxic properties in vitro; and the third one to demonstrate the utility of the ECF for direct nerve stimulation in an in vivo rodent model.

## 1. Introduction

Electrodes are used to stimulate and record electrical activity from the mammalian central nervous system and have been widely applied in the medical field for the investigation of spinal cord injuries, strokes, sensory deficits, and neurological disorders. Neuromodulation, an evolving therapeutic approach based on the recording and delivery of electrical signals to targeted neurological sites, has been investigated for the treatment of conditions [1,2,3,4,5,6,7,8] such as Parkinson’s disease [9,10], dystonia, [11,12], tremor [13], psychiatric conditions [14], and pain disorders [15,16,17,18]. Most neural recording and stimulating electrodes are metallic [19]. While metallic electrodes are highly efficient conductors of electric charge, they present several disadvantages when interfacing with neural tissue [20,21,22]. First, poor biocompatibility causes reactive gliotic changes in neural tissue [23,24,25]. Second, metals used as current injectors (e.g., platinum and iridium) are expensive and fragile. Third, the resistance inherent in metallic conductors causes energy to be dissipated as heat, which can worsen reactive changes in the surrounding tissues [26,27,28], thus decreasing electrode efficiency and potentially damaging biological tissue.

Next-generation electrodes are being designed to eliminate or minimize the abovementioned concerns without greatly affecting sensitivity [29,30,31]. Such electrodes are based on material substrates with either an organic composition or a modified metallic structure [32,33]. These electrodes can both capture and deliver adequate electrical currents while promoting neural integration with reduced tissue injury and inflammation [34,35,36,37,38]. Carbon fiber-based electrodes are one such alternative and have been shown to serve as effective measurement and stimulation tools [39,40]. In some cases, carbon fibers have been attached to silk fiber to reduce tissue damage, decrease electrode stiffness, and improve recording stability [41]. Electrodes coated with nanoparticles have also shown promise for neurological monitoring and treatment. In animal models, nanoparticle coating reduces gliosis while improving contact impedance [42].

We have previously reported on electrically conductive organic fibers (ECFs) which utilize a novel conductive ink [43] composed of single-walled nanotubes and regioregular poly(3-hexylthiophene) stained on nonconductive cotton, polyester, or silk fibers. In our previous paper, we both demonstrated that biosignals could be recorded with the stained polyester fibers and reported on a number of the characteristics of the fibers. Furthermore, the fibers have been utilized to fabricate the smart wearable multisensing textile-based system for real-time prediction of disease onsets. The system is an extension of our previous reported projects [44,45,46]. In this paper, we demonstrate that the polyester ECFs may be used for electrical stimulation of neural tissue in an in vivo rodent model. In addition, the electrical conductivity properties of the ECF and the cytotoxicity of the fiber cultured in human cells are investigated. The organization of the paper is as follows: we first describe the methods utilized for the fabrication of the ECFs and characterization of their electrical properties; then, cytotoxicity studies; and finally, stimulation studies in an animal model and statistical analysis. We report the results of these studies. The results are then discussed, and our conclusions are summarized.

## 2. Methods

### 2.1. Fabrication of Electrically Conductive Organic Fibers (ECFs)

Fibers based on commercial polyester threads were prepared. Off-the-shelf polyester threads (Joann Fabrics and Crafts) were coated with conductive ink without any additional chemical treatment. To prepare the conductive ink, regioregular poly(3-hexylthiophene) (rr-P3HT, 98% purity) was purchased from American Dye Source, Inc. (Montreal, QC, Canada) and single-walled carbon nanotubes (SWCNTs ≥95% carbon basis ≥99%, 0.84 nm average diameter) were purchased from Sigma Aldrich (St. Louis, MS, USA), and the SWCNTs were added to the rr-P3HT solution in CHCl3, as previously described [43]. The resulting mixture was ultrahigh sonicated using a Microson Ultrasonic Cell Disruptor for 30 min in an ice bath to avoid overheating and undesirable secondary reactions. A dipping and drying technique was utilized to stain the polyester fibers with the conductive ink. The process was repeated ten times per ECF. After staining, the fibers were oven dried for 15 min at 100 °C.

### 2.2. Electrical Impedance

The electrical impedance of the fibers as a function of both frequency and length were measured. In the first set of measurements, three polyester ECFs of the same length (l = 2.5 cm) were obtained from one 7.5 cm fiber. The resistance of each fiber was measured using a voltage-ohmmeter (VOM) (Fluke 87-V) and an impedance analyzer (Agilent 4395A) at 1 MHz. In the second set of measurements, the impedance of each 2.5 cm ECF was also measured over the 100 kHz–100 MHz (100,000 kHz) frequency range using the same impedance analyzer. In the third set of measurements, the resistance (R) and reactance (X) as a function of fiber length were measured. In these measurements, a 4 cm sample was cut from a 12 cm fiber thread and the resistance and reactance values were measured for five trials with the impedance analyzer at 1 MHz. The 4 cm thread was then reduced in length by 0.5 cm, and the measurements were repeated. This procedure was repeated to the shortest length of 2 cm.

### 2.3. Cell Culture and Cytotoxicity

The loss of membrane integrity of three human cell types cultured with the ECFs was investigated. Human embryonic kidney cells (HEK293; ATCC^®^ CRL-1573™) (American Tissue Culture Collection, Manassas, VA, USA), murine embryonic fibroblast cells (NIH3T3; ATCC^®^ CRL-1658™) (American Tissue Culture Collection, Manassas, VA, USA), and human keratinocyte cells (HaCaT; #T0020001, AddexBio, San Diego, CA, USA) were grown and maintained in Dulbecco’s Modified Eagle Medium (DMEM; Gibco, Gaithersburg, MD, USA), supplemented with 10% fetal bovine serum (Thermo Fisher Scientific, Waltham, MA, USA) and 1% penicillin-streptomycin (Gibco, Gaithersburg, MD, USA). Then, 1 × 10^5^ cells were plated in the wells of three 12-well microplates (Corning Inc., Corning, NY, USA), one for each cell type, and allowed to adhere overnight. Three 2 mm length polyester ECFs were added to nine treatment wells for each microplate and co-incubated with cells for 72 h without media change. Three control (no ECF) wells were also analyzed for each cell type. Post incubation, the ECFs were removed, and cells were labeled with CyQUANT^®^, a nucleic acid-sensitive fluorescence assay (Thermo Fisher Scientific, Waltham, MA, USA). Direct fluorescence intensities were measured with a spectrophotometer reader (TECAN, Männedorf, Switzerland). Cytotoxicity was assessed by normalizing the fluorescence intensities to the control group (no treatment) and plotted as percent viability.

### 2.4. Animal Model

The animal experiment was performed under the Allegheny General Hospital Institutional Animal Care and Use Committee approval (IACUC approval number 1036). A male Sprague Dawley rat (110 weeks old, 1.5 kg) was utilized. The rat was anesthetized with ketamine (90 mg/kg) (Mylan Pharmaceuticals, Morgantown, WV, USA). Maintenance anesthesia was achieved using isoflurane (Piramal Critical Care Bethlehem, Pennsylvania, USA) with 4% flow delivered via face mask. Thermoregulation was provided with a heating pad. The depth of anesthesia was monitored with response to toe pinch stimulus and bradypnea. The area over the left groin was clipped and prepped using iodine. Sharp dissection was used to expose the neurovascular bundle in the groin. The sciatic nerve was carefully separated from vascular structures with a sectioned vascular loop. Standard metal subdermal needle electrodes (Ambu Neuroline subdermal needle electrode 12 × 0.40 mm, Ballerup, Denmark) were utilized as described below. A metal ground electrode was placed in the subcutaneous tissue of the left lower quadrant. For initial stimulation, a metal anode electrode was placed medial to ground in the subcutaneous tissue of the left lower quadrant and a metal cathode electrode was directly applied to the dissected sciatic nerve and hand held in place. Metal recording electrodes were placed in the left tibialis anterior muscle in bipolar montage to record the evoked compound muscle action potentials (CMAP). All stimulation and recordings were performed using a NeuroNet VI system (Computational Diagnostics Inc., Pittsburgh, PA, USA).

Stimulus frequency was set at 5.1 Hz and pulse width was set at 200 µs. The sciatic nerve was stimulated with current starting at 0 mA and increased in 0.1 mA increments until CMAPs were observed at 0.9 mA, and data were recorded at 1 mA. In the second experiment, the metal cathode was replaced with a polyester ECF (l = 10 cm) by gently wrapping the flexible ECF around the nerve, avoiding contact with any other tissue. The stimulation parameters used previously were then repeated. The threshold was again reached at 0.9 mA, and CMAPs were recorded at 1 mA. In the final experiment, the metal anode electrode was replaced by a polyester suture-based ECF (l = 10 cm) sutured directly into the fascia medial to the exposed sciatic nerve, resulting in a stimulating setup consisting of an ECF anode and cathode. The previously described stimulation parameters were repeated with the following exception: the CMAP threshold was found at 2.1 mA and CMAPs were subsequently recorded at a stimulus intensity of 2.2 mA, followed by recording at 2.3 mA. For all ECF stimulation experiments, the coated polyester fibers utilized as either stimulating anodes or cathodes were attached to Grass 10mm gold cup electrode (Natus, Middleton, WI, USA) with Ten20 conductive paste (Weaver, Aurora, CO, USA) to provide connection to the stimulator. The nerve was rested between experiments. Upon completion of the study, the rat was euthanized using carbon dioxide at a rate of 40% chamber volume per minute, followed by bilateral thoracotomy in a procedure approved by the American Veterinary Medicine Association.

### 2.5. Statistical Analysis

All data were analyzed with either a Student’s *t*-test or analysis of variance (ANOVA), followed by Tukey’s post hoc tests for multiple comparisons between treatment groups and their controls using Prism 8 (GraphPad, San Diego, CA, USA). Statistical significance was defined at *p* ≤ 0.05.

## 3. Results

### 3.1. Fiber Characterization

The polyester ECF fibers were observed with a light microscope, as shown in Figure 1. Using vernier calipers, a fiber diameter of 200–250 µm was measured for each ECF.

### 3.2. Electrical Measurements of Polyester ECFs

The resistance and the resistive component of the impedance for each of the three 2.5 cm ECF samples measured as described in the Methods section are shown in Table 1.

In the second experiment, the resistive impedance of the polyester ECFs was measured as a function of frequency over the range of 100 kHz–100 MHz. This frequency range was chosen to fully capture the changes in the resistive impedance. As shown in Figure 2, when the frequency applied to the sample exceeded 5 MHz (approximately at the −3 dB point), the sample resistive impedance decreased at the rate of approximately 20 dB/dec.

In the third experiment, the resistance and reactance values as a function of sample length were measured for five trials with the impedance analyzer at a frequency of 1 MHz. Figure 3 shows box plots of the resistance values of the polyester fibers as a function of the sample length. The resistance increased as a function of the length of the polyester fibers.

Figure 4 shows the box plot of the reactance values of polyester fibers. This data is consistent with a capacitive reactance with capacitance value of 88 picofarads at 4 cm length and 270 picofarads at 2 cm length.

### 3.3. Fiber Cytotoxicity Testing

We found no statistically significant impact of incubation with coated ECFs in any of the cell lines compared to the control wells (Figure 5), and all wells exhibited essentially nearly 100% viability. These data suggest that in all three cell lines tested, there was minimal toxicity from the ECFs in vitro for the time period tested.

### 3.4. In Vivo Stimulation

We compared the capability of the ECFs to provide stimulation in vivo to stimulation provided by a metal cathode and anode. The electrode was placed onto the nerve in the similar manner to the fiber. The amount of steel and fiber was consistent across material tested. The individual waveforms are presented in Figure 6 and Figure 7. Both the steel and polyester ECF were placed on top of the exposed nerve fiber. In the first experiment, the sciatic nerve was stimulated with a metal cathode and a metal anode with current starting at 0 mA and increased by 0.1 mA increments until a compound muscle action potential (CMAP) was recorded by a pair of subdermal recording electrodes placed in the left tibialis anterior muscle. The threshold was determined to be 0.9 mA and the CMAPs were then recorded at 1 mA, as shown in Figure 6.

In the second experiment, the metal cathode was replaced with an ECF. The fiber was wrapped around the sciatic nerve, avoiding contact with any other tissue. The sciatic nerve was again stimulated starting at 0 mA and increased by 0.1 mA increments until CMAPs were obtained. The threshold was again reached at 0.9mA, with the CMAPs (Figure 7) recorded at 1 mA stimulation.

In the third experiment, the metal anode was replaced by an ECF sutured directly into the fascia medial to the exposed sciatic nerve. The sciatic nerve was again stimulated with current starting at 0 mA and increased by 0.1 mA increments until CMAPs were observed in the metal recording electrodes. In this case, the threshold was at 2.1 mA, and CMAPs were recorded at 2.2 mA and 2.3 mA (Figure 8).

## 4. Discussions

In a previous study [43], we demonstrated that polyester ECFs could be used to record bioelectric data, and we measured the resistance of the polyester ECFs as approximately 650 ohms per cm, while the electrode–skin impedances were measured to be approximately between 18 and 38 kohms at 400 Hz. This study was designed to demonstrate that we could also stimulate neural tissue using the polyester ECFs and to further define their impedance characteristics, as well as demonstrate that they would have no cytotoxic effect for short-term exposure to human cell lines.

The resistance (DC) and resistance impedance (at 1 MHz) values of the polyester ECF were measured to be between 2140 and 2500 ohms/cm in three 2.5 cm samples. This was about four times the values measured previously [43]. We believe the variances in the resistance measurements are related to variations in the structure of the ECFs due to the current fabrication process which does not control well for diameter of the coated fiber or the degree of saturation of the fiber by the conductive ink.

The resistive component of the impedance for the same three samples referenced above was measured between 100 kHz and 100 MHz. The values were approximately constant out to approximately 5 MHz, and then decreased at 20 dB/dec. This demonstrates that over the frequency range from 5 MHz to 100 MHz, it was increasingly easy for charge to flow through the fibers. This is not easy to explain. However, this could be consistent with the inverse of the well-known skin effect. At lower frequencies, the charge flows along the surface of the ECFs, and as the frequency increases, the charge flows deeper in the material. This is a new concept. These results suggest that we need to develop a deeper understanding of the chemical structure of these coated fibers and how electrons are moved between the constituent molecules.

In the third resistance/impedance experiment, we measured the resistance and the reactance of the ECF as a function of length at 1 MHz. Each measurement was repeated five times on the same sample, demonstrating a high accuracy to the measurement technique. The resistance increased as a function of length, as expected. The reactance was negative and capacitive in nature, and also increased as a function of length going from 0.6 kohms at 2 cm in length to 1.8 kohms at 4 cm in length. This is consistent with capacitors connected in series where the reciprocal of the equivalent capacitance equals the sum of the reciprocals of the individual capacitances.

We also investigated the cytotoxicity of the polyester ECFs, in a 72 h experiment, using three human cell lines. There was no cytotoxicity found over that period of time for those particular cells. This does not fully answer the question of cytotoxicity or the possibility of a gliotic reaction in vivo but gives us encouragement to move forward with more detailed in vivo experiments.

Finally, we investigated the ability of the polyester ECFs to function as a stimulating electrode. In these experiments, we demonstrated that the ECF could, indeed, be used to stimulate neural tissue. The major distinction in these experiments is that the ECF–ECF electrodes required 230% more current to produce an equivalent response, compared to the experiments in which the electrodes were metal. This may be related to the higher resistance impedance of the ECF compared to metal electrodes. These results again suggest that a deeper understanding of the charge flow within these conductive ink-coated fibers is needed.

## 5. Conclusions

We have proposed a polyester-based organic conductive sensing fiber and the coating process with the biocompatible conductive ink for in vivo neural tissue stimulation and recording. We demonstrated that these fibers may perform both stimulation and recording. Preliminary characterizations of resistance and impedance of the fibers as a function of both length and frequency have been investigated. In another preliminary experiment, we demonstrated that the polyester ECFs are not cytotoxic against three different human cell types. Thus, it demonstrates the suitability of the fiber for further investigation into its usefulness as interfaces to neural tissue. Future work will involve the development of conductive materials for lead, wires, electrodes, and interconnections that enhance the functionality, wearability, and reusability of electronics and biosensors.

## Figures and Tables

**Figure 1 sensors-21-07581-f001:**
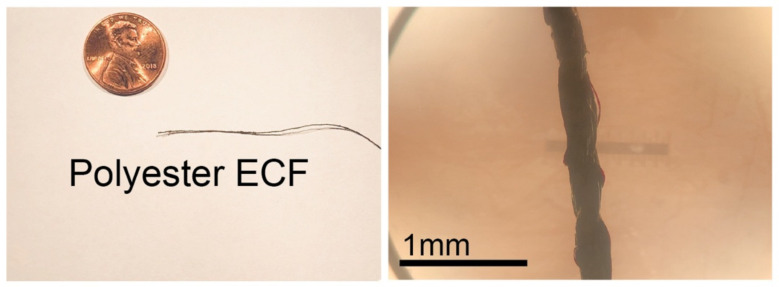
Macro (**left**) and microscopic image (**right**) of polyester-based ECF.

**Figure 2 sensors-21-07581-f002:**
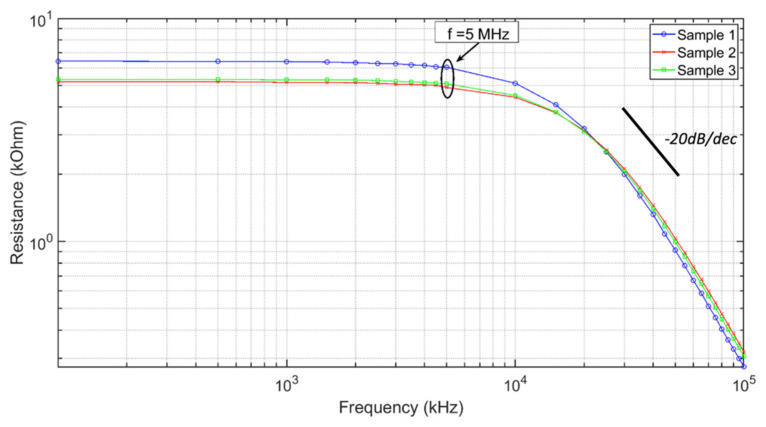
Resistance component of the impedance of three 2.5 cm long polyester samples taken from the same 7.5 cm ECF, measured with the impedance analyzer. The resistance decreased as a function of frequency above 5 MHz at a rate of approximately 20 dB/dec.

**Figure 3 sensors-21-07581-f003:**
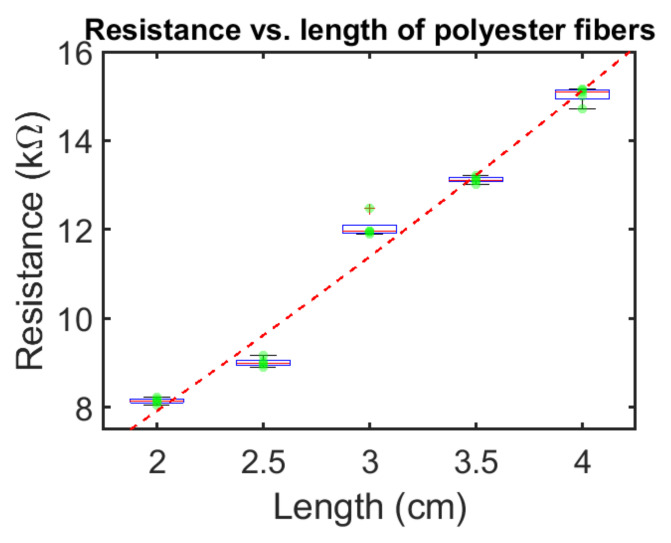
Resistance values of the polyester ECFs measured at f = 1 MHz, five times, with respect to the fiber length. Box plot with box of first and third quartile and median, and whiskers of minimum and maximum values, neglecting outliers. Outliers are denoted as “+”.

**Figure 4 sensors-21-07581-f004:**
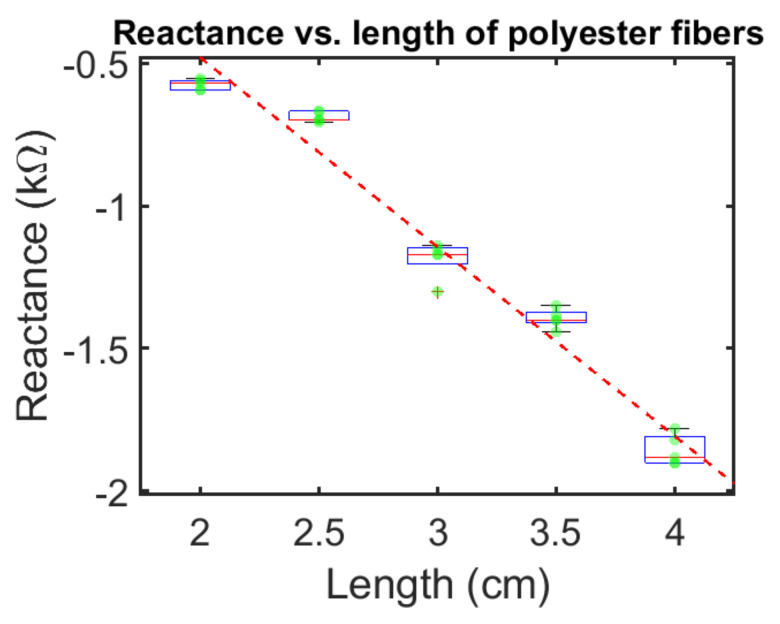
Reactance of polyester ECF measured at f = 1 MHz, five successive times, respectively, with respect to the fiber length. Box plot with box of first and third quartile and median, and whiskers of minimum and maximum values, not considering outliers. Outliers are denoted as “+”.

**Figure 5 sensors-21-07581-f005:**
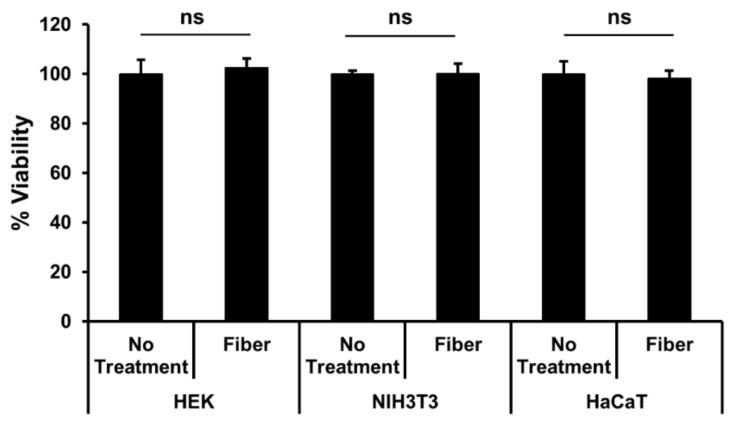
Effect of ECF fibers on the viability of HEK293, NIH3T3, and HaCaT cells. Cells were incubated in the presence of 2 mm length ECF fibers, and the viability was determined by the direct CyQUANT™ assay at 72 h. Cells grown on tissue culture plastic were used as the negative control (no treatment group). Results are expressed as a percentage of negative control and bars indicate mean ± SEM (n = 3 wells for each group), ns = not significantly different.

**Figure 6 sensors-21-07581-f006:**
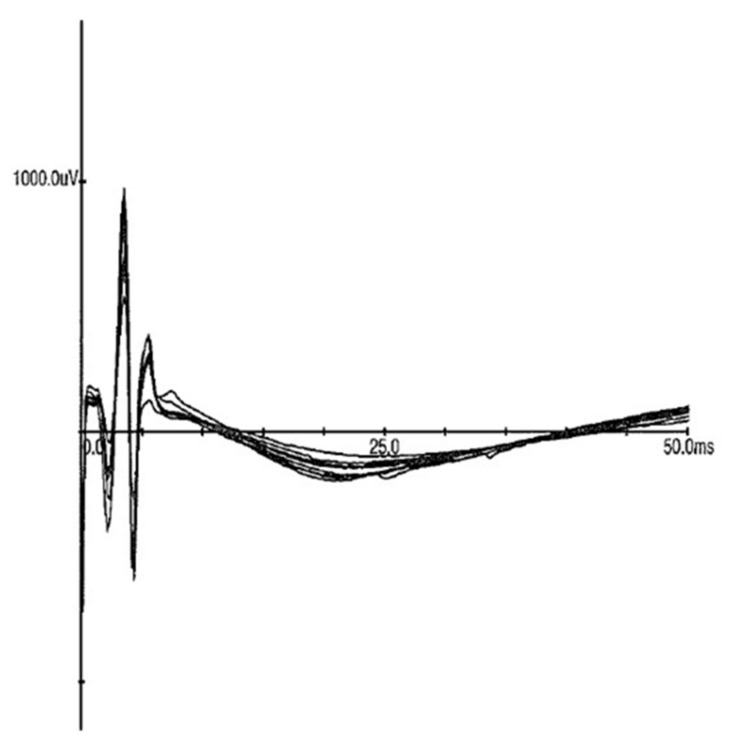
Stimulation provided through metal electrodes with the cathode handheld against the nerve, producing CMAPs recorded with subdermal needle electrodes at 1 mA stimulating current. Eight successive CMAPS are overlaid.

**Figure 7 sensors-21-07581-f007:**
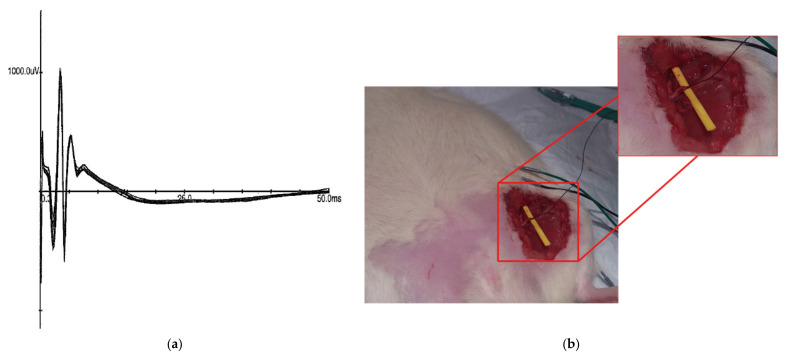
Metal–ECF stimulation configuration of sciatic nerve. Panel (**a**) presents 10 successive CMAPs overlaid, obtained by stimulation of the sciatic nerve using an ECF as a cathode at 1 mA. Panel (**b**) shows the ECF looped around sciatic nerve propped up with a piece of nonconductive yellow tubing.

**Figure 8 sensors-21-07581-f008:**
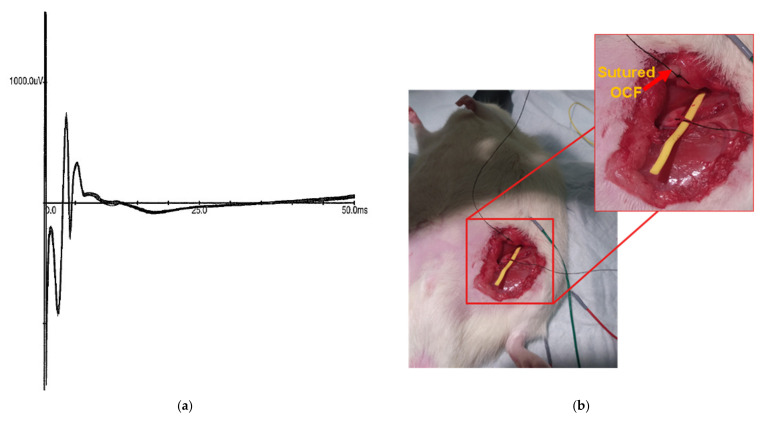
ECF–ECF stimulation configuration of sciatic nerve. Panel (**a**) presents 10 successive CMAPs overlaid, obtained by stimulation of the sciatic nerve using an ECF as a cathode and anode at 2.3 mA. Panel (**b**) shows the sutured anodal ECF (red arrow) adjacent to the sciatic nerve with the cathodal ECF looped around sciatic nerve, propped up with a piece of nonconductive yellow tubing.

**Table 1 sensors-21-07581-t001:** Resistance values obtained from three polyester ECFs, each of 2.5 cm length using a voltage-ohmmeter (center column) and the impedance analyzer (right column).

ECF Sample	Resistance (DC)	Resistance (at 1 MHz)
1	6.40 kohms	6.48 kohms
2	5.16 kohms	5.19 kohms
3	5.35 kohms	5.35 kohms

## Data Availability

Not applicable.

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
