# Peer review of "Development and Characterization of Novel Conductive Sensing Fibers for In Vivo Nerve Stimulation"

_sensors, 2021, doi:10.3390/s21227581_

Round 1
Reviewer 1 Report
lines 150-152: "Upon completion of the study, the rat was euthanized according to the procedure approved by the American Veterinary Medicine Association. "
Please describe exactly euthanized method used.
Author Response
lines 150-152: "Upon completion of the study, the rat was euthanized according to the procedure approved by the American Veterinary Medicine Association. “Please describe the exactly euthanized method used.
We thank the reviewers for the suggestions, description of the procedures has been added to the manuscript.” Rat was euthanized using carbon dioxide at a rate of 40% chamber volume per minute followed by bilateral thoracotomy”. These statements have been added in the last paragraph of session 2.4

Reviewer 2 Report
In this work, the authors prepared an organic conductive sensing fibers consisting of single walled carbon nanotubes and poly(3-hexylthiophene) for in vivo Nerve Stimulation. The work could be published after addressing the following issues:
- It's better to provide the purity of chemicals. Have they done any pre-treatment?
- In page 4, For Figure 1, they mentioned that in the manuscript, they have done the scanning electron microscope. However, I see Figure 1 is macro and micro photographs. Please indicate them clearly.
- In page 7, line 5th, it should be " reactance values are decreased as the length is increased".
- There are some references that may help improve the paper: Carbon, 142, (2019) 131-140; ACS Appl. Mater. Interfaces 2018, 10, 26713−26722.
Reviewer 3 Report
The main objective of the paper seems to be to demonstrate the feasibility of using off-the-shelf fibers coated with carbon nanotubes to stimulate nerves. Overall, the idea of conductive, flexible leads for stimulators reflects a need in the current systems for better contacting neural interfaces for improved efficacy. While the authors present promising data on the stimulation capabilities, the capability of these conductive fibers to sense or record a signal is unclear in this work. It is a significant issue in the neural interface field to do both stimulation/recording or sensing using the same electrode and suggest that the title be changed to remove ‘sensing’.
- The stability of the conductive nanotube ink is unclear under invitro/in vivo conditions.
- In addition to the cytotoxity tests, did the authors perform a soak test to show the stability of the electrode characteristics?
- Also, in vivo the nerve may undergo motion that may cause the coated suture to chafe and potentially change electrode impedance. Did the authors test the reliability of the electrode impedance characteristics before and after stimulation?
- What is the charge injection capacity of these electrodes?
- For Tables 1 & 2 what frequency was used from the impedance analyzer to compare to the volt/ohmmeter?
- It is interesting that the CMAP threshold increases for the OCF-OCF (Ln145-146). Could this be due to potential constriction of the nerve by the suture? Ln344-345 also mentions the increase in threshold. Can the authors discuss why this might be the case (poor contact? Nerve constriction? Etc.)
- The placement of the electrodes is ambiguous for the Steel-steel (Figure 7). An image showing position of the electrodes similar to figures 8&9 would be helpful. How far apart were the anode and cathode placed on the nerve and was this consistent across the different combinations of electrodes tested? Also, what were the relative contact area of the electrodes on the nerve?
- Why is there no latency in the CMAP from the stimulus onset in Figure 7 compared to Figure 8 & 9? Also, are these CMAP waveform averaged over multiple stimulations? If so, please describe in methods the signal analysis parameters.
- In Figure 10, did the authors vary the order of the stimulation combinations? Aside from potentially better contact between nerve and fiber (Ln 296-298), could entrainment have played a role in the tighter distribution of CMAP amplitudes? Additional experiments are suggested to bolster this claim.
Minor issues:
- Ln 105: Typo: 1 x 105 cells?
- Ln 254 Typo: mAA
Round 2
Reviewer 3 Report
The revision is much better focused and the manuscript is acceptable for publication.